# Biosensors-Based In Vivo Quantification of 2-Oxoglutarate in Cyanobacteria and Proteobacteria

**DOI:** 10.3390/life8040051

**Published:** 2018-10-27

**Authors:** Hai-Lin Chen, Amel Latifi, Cheng-Cai Zhang, Christophe Sébastien Bernard

**Affiliations:** Aix Marseille Univ, CNRS, LCB, IMM, 13009 Marseille, France; c_hailin@yahoo.fr (H.-L.C.); latifi@imm.cnrs.fr (A.L.); cczhang@imm.cnrs.fr (C.-C.Z.)

**Keywords:** biosensor, cyanobacteria, *Escherichia coli*, FRET, 2-oxoglutarate

## Abstract

2-oxoglutarate (α-ketoglutarate; 2-OG) is an intermediate of the Krebs cycle, and constitutes the carbon skeleton for nitrogen assimilation and the synthesis of a variety of compounds. In addition to being an important metabolite, 2-OG is a signaling molecule with a broad regulatory repertoire in a variety of organisms, including plants, animals, and bacteria. Although challenging, measuring the levels and variations of metabolic signals in vivo is critical to better understand how cells control specific processes. To measure cellular 2-OG concentrations and dynamics, we designed a set of biosensors based on the fluorescence resonance energy transfer (FRET) technology that can be used in vivo in different organisms. For this purpose, we took advantage of the conformational changes of two cyanobacterial proteins induced by 2-OG binding. We show that these biosensors responded immediately and specifically to different 2-OG levels, and hence allowed to measure 2-OG variations in function of environmental modifications in the proteobacterium *Escherichia coli* and in the cyanobacterium *Anabaena* sp. PCC 7120. Our results pave the way to study 2-OG dynamics at the cellular level in uni- and multi-cellular organisms.

## 1. Introduction

Within a cell, the coordination of distinct metabolic pathways is mediated by fluctuations in the concentrations of specific metabolites, allowing the allosteric transfer of this information to sensor proteins, thereby generating the appropriated metabolic response. Metabolites that act as key players in microbial signal transduction include glutamine, 2-oxoglutarate (α-ketoglutarate, 2-OG), cAMP, and ppGpp [1]. Each of these metabolites is dedicated to the transfer of information upon the availability of a specific class of nutrient.

2-OG is an intermediate metabolite of the tricarboxylic acid (TCA or Krebs) cycle and is used as a carbon skeleton for nitrogen assimilation. It therefore coordinates carbon and nitrogen metabolisms. The TCA cycle is an essential metabolic pathway conserved in all living organisms and provides not only precursors for biosynthesis of a large number of biomolecules, but also reducing power for the production of the biological form of energy, ATP, through respiration. The 2-OG level reflects also the carbon and nitrogen balance, a signaling characteristic that is conserved throughout organisms. An inverse correlation between ammonium availability and 2-OG accumulation has been demonstrated in various microorganisms such as *Escherichia coli*, archaea, cyanobacteria, and budding yeast [2,3,4]. The signaling role of 2-OG is well described in cyanobacteria and especially in the filamentous heterocystous cyanobacterium *Anabaena* sp. strain PCC 7120 (hereafter *Anabaena*). In *Anabaena*, combined-nitrogen depletion leads to the accumulation of 2-OG, which acts as a trigger to elicit heterocyst differentiation [3,5,6]. Heterocysts are cells specialized in nitrogen fixation, which differentiate from vegetative cells. Heterocysts and vegetative cells are interdependent: heterocysts provide fixed nitrogen from atmospheric N_2_ in the form of amino acids to vegetative cells, while the latter use photosynthesis to fix carbon from atmospheric CO_2_ and provide sucrose to heterocysts [5,7,8]. For its function as signaling molecule, 2-OG binds to the regulatory proteins, PII and NtcA. The crystal structures of these two regulators, alone or complexed to 2-OG, showed that they undergo significant conformational changes when binding to their ligands [9,10,11,12,13]. PII, encoded by the *glnB* gene, is conserved in bacteria and plants. It assembles as homotrimer and performs its regulatory function through protein–protein interactions [14]. In cyanobacteria, the activity of PII is regulated by phosphorylation, and by the binding of effectors such as ATP and 2-OG, enabling PII to adopt different conformations, and to interact with different protein partners under different conditions [15]. In *Anabaena*, the PII trimer binds to 3 molecules of PipX in conditions of combined nitrogen deprivation. Interestingly, PipX and 2-OG share the same binding site on PII, and are therefore mutually exclusive. When the ammonium cell concentration decreases, 2-OG level increases and chases PipX from PII. In addition, PipX is a co-activator of the cyanobacterial-specific transcriptional factor NtcA which belongs to the CRP/CAP family of homodimeric transcriptional regulators, and uses 2-OG as the inducer molecule [16,17]. One molecule of 2-OG binds to each monomer of the NtcA dimer in a pocket located in the N-terminal region. 2-OG binding on NtcA reduces the distance between the two DNA-binding F-helixes of the NtcA dimer from 37 Å in the apo form to 34 Å in the 2-OG-NtcA complex, an optimal distance for DNA binding [3,9]. The activity of NtcA is also controlled by its interaction with PipX. Each NtcA subunit binds one subunit of PipX in the complex and this interaction is required for NtcA-mediated gene expression. It has been proposed that PipX stabilizes the active form of NtcA and helps NtcA to recruit the RNA polymerase [4,10,16,17,18]. In *Anabaena,* an *ntcA* mutant is unable to initiate heterocyst differentiation [5,19]. Based on the observation that 2-OG binds to NtcA and that an increase in the levels of NtcA and 2-OG triggers heterocyst differentiation, it has been proposed that the interaction between 2-OG and NtcA constitutes the very first step in heterocyst development [3,5]. However, the role of NtcA is not limited to the initiation of heterocyst development, as it has been shown to have a broad regulatory function. The NtcA regulon comprises several hundreds of genes, including targets involved in both carbon and nitrogen metabolism, stress response, and in different steps of heterocyst differentiation and functioning [5,20].

In addition to PII and NtcA, 2-OG levels also modulate the activity of many regulatory proteins and pathways, predominantly related to nitrogen or carbon metabolism in bacteria, archaea, algae, plants and animals [1,11,15,21,22,23]. Despite the central role of 2-OG and its extensive regulatory repertoire, we still lack tools to appropriately measure its concentrations and dynamics in vivo. Enzymatic assays, mass spectrometry or high-pressure liquid chromatography (HPLC) approaches have been developed to estimate 2-OG levels in cell extracts [4,24,25,26,27,28,29,30]. However, 2-OG such as many metabolites, turns over very quickly and may be lost or degraded during extraction. As a consequence, 2-OG concentration measurements are non-reproducible and underestimated [30,31,32]. In addition, none of these approaches allow to follow 2-OG dynamics. To circumvent these disadvantages, 2-OG biosensors based on fluorescence resonance energy transfer (FRET) have been recently developed. FRET relies on the distance-dependent interaction between two fluorescent molecules, and on the transfer of the excited state of the donor molecule to the acceptor when these two molecules are in close proximity [33,34]. We have constructed a 2-OG biosensor based on the interaction between the *Anabaena* PII and PipX proteins, whose interaction is disrupted by 2-OG binding to PII. The FRET ratio (fluorescence of the acceptor YFP/fluorescence of the donor CFP) has been shown to be inversely proportional to 2-OG levels in vitro in a 0.01–10 mM range [35]. A second biosensor, based on the 2-OG-dependent interaction between the *Synechococcus elongatus* PII and *N*-acetyl-glutamate kinase (NAGK) proteins, has been shown to measure 2-OG levels in the 0.01–1 mM range [36]. Although such biosensors have been proved to be highly efficient in vitro or ex vivo, they failed to measure 2-OG variations within living cells [35,36,37]. One of the limitations is that it was difficult to achieve similar levels of production for the two fusion proteins in *E. coli* and *Anabaena*. For this reason, 2-OG biosensors based on single proteins need to be developed to measure 2-OG levels in vitro and in vivo. In this study, we report the engineering of two new biosensors, based on 2-OG-dependent conformational changes of PII or NtcA. We show that they could both report 2-OG changes, with a different concentration range in vitro and in vivo in two bacteria belonging to separate phyla.

## 2. Materials and Methods 

### 2.1. Strains and Growth Conditions

*E. coli* K-12 strains DH5α and BL21(DE3) (Appendix A) were used for construction of recombinant plasmids, and protein production, respectively. *E. coli* strains were routinely grown in Lysogeny Broth or M9 minimal media with antibiotics when required (ampicillin, 100 μg/mL; kanamycin, 100 μg/mL). Unless specified, *Anabaena* sp. PCC7120 strains were grown at 28 °C in BG11 medium (containing sodium nitrate as the source of combined nitrogen) or BG11_0_ medium (BG11 medium without sodium nitrate) with an illumination of approximately 30 μmol photons m^−2^s^−1^, supplemented with antibiotics when required (streptomycin, 2.5 μg/mL; spectinomycin dihydrochloride pentahydrate, 2.5 μg/mL). Conjugation in *Anabaena* was performed as previously described [38] and ex-conjugants were verified by polymerase chain reactions (PCR) and DNA sequencing.

### 2.2. Plasmid Constructions

Plasmids and oligonucleotides used in this study are listed in Appendix A. PCR was performed with a Biorad thermocycler, using the PrimeSTAR DNA polymerase (Ozyme). Plasmids pET-15b-F_NtcA_1 to 10 were constructed by ligation of *Spe*I-*Kpn*I fragments of the *Anabaena ntcA* gene into pFRET12aa [35,39]. For construction of plasmid pET-15b-F_PII, a *Xho*I restriction site was first introduced by site-directed mutagenesis after *glnB* codon 48 (Gly-48 position in the PII T-loop) into plasmid pET-PII_CYPet [35] using oligonucleotides glnB_loop_XhoI_F and glnB_loop_XhoI_R, to yield pET-PII(X)-CYPet. The *yfp* gene was then amplified from pYPet [39] using oligonucleotides YFP_XhoI_F and YFP_XhoI_R. The primers introduce *Xho*I sites at both extremities, allowing in-frame insertion of *yfp* downstream *glnB* codon 47. Two new vectors carrying mTurquoise instead of CFP in pET-15b-F_NtcA_2 and pET-15b-F_PII were engineered by amplification of the *m-turquoise* gene from pmTurquoise2-C1 [40] using oligonucleotides mTurquoise-KpnI_F and mTurquoise_NotI_R, and insertion into the *Kpn*I-*BamH*I destination vectors, to yield pET-15b-F-mT_PII and pET-15b-F-mT_NtcA. Point mutations were introduced by site-directed mutagenesis using complementary pairs of oligonucleotides bearing the desired mutation (glnB_K58M_F and glnB_K58M_R for PII K58M, and ntcA_R88E_R and ntcA_R88E_F for NtcA R88E) into pET-15b-F-mT_PII and pET-15b-F-mT_NtcA to yield pET-15b-F-mT_PII^K58M^ and pET-15b-F-mT_NtcA^R88E^, respectively. The integrative vector pRL278 [41] was used for the integration of exogenous sequence in a neutral site (from *alr7261* to *all7262*) located in the alpha mega plasmid in *Anabaena* as described [42]. For construction of pRL278-P*_petE_*_F-mT_PII and pRL278-P*_petE_*_F-mT_PII^K58M^, a *Xho*I-*Not*I P*_petE_* promoter fragment amplified from *Anabaena* DNA using primers PpetE_SalI_F and PpetE_MCS_NotI_R was first ligated into pRL278 (digested by *Xho*I-*Not*I) to yield pRL278-P*_petE_*-MCS. *Spe*I-*Not*I F-PII-mT and F-mT_PII^K58M^. Fragments were then amplified from pET-15b-F-mT_PII and pET-15b-F-mT_PII^K58M^ respectively using Ppet_SpeI_F and T7_ter, and inserted into pRL278-P_petE_-MCS to generate pRL278-P*_petE_*_F-mT_PII and pRL278-P*_petE_*_F-mT_PII^K58M^. All PCR products and constructs were verified by DNA sequencing.

### 2.3. Production of the FRET Fusion Reporters in E. coli and Anabaena for In Vivo Studies

*E. coli* BL21(DE3) cells carrying pET derivative plasmids were grown in M9 minimal medium (0.4% glucose, 10 mM NH_4_^+^) to optical density at λ = 600 nm (OD_600_) of 0.5 and gene expression was induced with 10 µM of isopropyl-β-thio-galactopyrannoside (IPTG) for 16 h at 20 °C. *Anabaena* recombinant strains bearing the chromosomal P_petE__F-mT_PII insertion were grown to OD_750_ of 0.2 in BG11 and gene expression was induced with 1 µM of Cu^2+^ for 48 h at 28 °C. For the combined nitrogen step-down experiments, *Anabaena* cells were centrifuged, washed twice with BG11_0_ medium, and grown in this medium. To control the nitrate step-down, we checked for formation of heterocysts after 24 h.

### 2.4. Protein Production, Purification and Spectroscopy

*E. coli* BL21(DE3) cells carrying pET derivative plasmids were grown at 37 °C in Lysogeny Broth to an OD_600_ of 0.5, and gene expression was induced by the addition of 100 µM IPTG for 12 h at 17 °C. Proteins were purified by immobilized metal ion chromatography as previously described [35]. Spectroscopic analyses were monitored in 96-well plates using a TECAN Infinite^®®^ 200 series microplate reader (Hännedorf, Switzerland) equipped with a dual injector head for kinetic measurements. The excitation and emission bandwidths of the instrument are 9 nm and 20 nm, respectively. For in vitro and in vivo experiments, fluorescence emissions were recorded using an excitation at 425 nm, and emissions at 530 nm and 480 nm for YFP and CFP, respectively. The FRET ratio was calculated as the 530/480 intensity ratio (YFPem/CFPem) and reflects the level of excitation of YFP by the CFP, which is directly linked to the distance between the two fluorophores. The relative FRET ratio was calculated as the FRET ratio for a specific 2-OG concentration relative to the same ratio without 2-OG. The data were analyzed by the PRISM 5 program (GraphPad Software, San Diego, CA, USA). For in vitro experiments, fluorescence emissions were recorded 5 min after ligand addition.

## 3. Results

### 3.1. Design of 2-Oxoglutarate Mono-Protein Biosensor Systems 

The structures of PII and NtcA in the apo form or in complex with 2-OG show that both proteins undergo structural transitions when bound to 2-OG. For PII, 2-OG binding triggers a conformational change in the flexible T-loop, which protrudes outward as an antenna [43]. When bound to 2-OG, the PII antenna moves closer to the C-terminal region of PII. We therefore chose to insert YFP within this loop, between residues Gly-48 and Ser-49, whereas the donor (CFP) was inserted at the C-terminus of PII (F-mT_PII construct, Figure 1A,C). For the NtcA homodimer, 2-OG binds to the effector binding domain (EBD) inducing a twist within the two C-helices and an increased interface between the two monomers [9]. To select a suitable biosensor for 2-OG with a high FRET-ratio change in function of 2-OG binding, we engineered 10 different biosensors with variable length (F-NtcA_1 to 10). All variants contain the 2-OG binding pocket, lack the DNA binding domain, and are fused to a N-terminal YFP and a C-terminal donor fluorophore. The different variants correspond to N-terminal truncations of the EBD domain (starting at residues 7, 24, 33, or 62) and C-terminal truncations after residue 143 (last residue of the C-helix) or 163 (upstream the DNA recognition helices) (Figure 1B,D).

### 3.2. Evaluation of Single-Protein Based FRET Sensors for 2-OG Measurements

We first tested whether the biosensors respond to 2-OG in vitro. While the YFP/CFP F-NtcA_1, _3, _4, and _7 to _10 biosensors were not produced or aggregated as inclusion bodies, the F-PII, F-NtcA_2, F-NtcA_5 and F-NtcA_6 were successfully purified (Appendix A). Their FRET ratio (YFPem/CFPem) was then measured in presence (1 mM) or absence of 2-OG. In agreement with the 2-OG-induced conformational change observed in the crystal structures, the FRET ratio of F_PII in presence of 2-OG increased by ~40% compared to the control without 2-OG. For the F_NtcA biosensors, only F-NtcA_2 presented significant increased FRET ratio upon addition of 2-OG compared to the control (Figure 1E). While F-NtcA_5 contains the shortest fragment of NtcA and F-NtcA_6 the longest, they both weakly responded to 2-OG. One possible explanation is that the two fluorophores are already at close proximity and the distance between the two is not significantly modified upon 2-OG binding. Based on these data, F_NtcA_2 was chosen for further studies and renamed F_NtcA. To engineer 2-OG biosensors usable for in vivo 2-OG measurements, less sensitive to photobleaching, and exploitable for corroborative intensity-based FRET experiments, we replaced CFP by mTurquoise. The two new biosensors named F-mT_PII and F-mT_NtcA were purified and tested in vitro. Measurements of the FRET ratio showed results comparable to that measured with CFP, with an increase of 20–30% upon addition of 2-OG compared to control conditions without 2-OG (Figure 1F,G). The increase of the FRET ratio for F-mT_PII and F-mT_NtcA was dependent on the 2-OG binding since PII and NtcA variants previously shown to be affected for 2-OG binding (K58M substitution in PII [12]; R88E substitution in NtcA [44]) did not show any increase in the FRET ratio compared to the controls (Figure 1F,G).

To gain further insights into the responsiveness of the two biosensors, a titration curve representing the FRET ratio as a function of 2-OG concentration was determined for different concentrations of the two mono-protein biosensors as previously described [35]. Varying the biosensor concentrations between 0.25 and 1 µM did not cause significant changes on the response of the two biosensors to 2-OG (Figure 2A,B). However, we noted that F-mT_PII and F-mT_NtcA responded to different ranges of 2-OG concentrations. F-mT_PII is ~10 times more sensitive to 2-OG than F-mT_NtcA with a K_d_ of 120 μM and a K_d_ of 1.9 mM, respectively. 

Hence, the F-mT_PII is suitable to measure levels of 2-OG ranging from 10 µM to 1 mM whereas F-mT_NtcA is suitable to measure 2-OG levels from 0.1 mM to 10 mM. The two biosensors thus offer the possibility to measure 2-OG levels in different hosts. Indeed, the internal pool of 2-OG varies from one microorganism to the other. The F-mT_NtcA is therefore adapted to bacteria with high 2-OG internal concentrations such as *E. coli* (~0.5 mM [45]) whereas F-mT_PII may be more suitable for bacteria with lower 2-OG internal concentrations such as *Anabaena* (~0.1 mM [3]).

Both biosensors tolerate a wide range of temperature (25–37 °C) (Figure 2E,F) and pH values (6.4–8.7) with little changes in sensitivity especially for F-mT_NtcA (Figure 2C,D). They show a high specificity toward 2-OG excepted F-mT_PII, which also responded to 2-MPA, a non-metabolizable analogue of 2-OG (Figure 2G) [46]. We then tested how fast the two purified biosensors respond to 2-OG changes. For this, we performed a time-lapse recording of the FRET ratio upon addition of 2-OG. Figure 3A,B show that F-mT_PII and F-mT_NtcA responded within a relatively short time. The measured half-times (time for reaching half of the FRET plateau) in presence of 1 mM of 2-OG were 44 s and 76 s for F-mT_PII and F-mT_NtcA, respectively. In addition, we observed that stable FRET ratios were maintained for long periods after the equilibrium. Overall, these in vitro results demonstrate that the two biosensors can respond rapidly to 2-OG changes.

### 3.3. The 2-OG Biosensors Are Responsive to 2-OG Changes In Vivo in Anabaena and E. coli

We finally tested whether the biosensors are sensitive enough to detect 2-OG variations in vivo. Based on the in vitro experiments described above, the F-mT_PII, that is more sensitive to low 2-OG concentrations, was used for studies in *Anabaena*. To avoid heterogeneous expression form replicative plasmids [47], the F-mT_PII and or F-mT_PII^K58M^ constructs were inserted into the chromosome of *Anabaena*. FRET was then measured over time after addition of dimethyl-2-OG, a membrane-permeable 2-OG analogue that is processed into 2-OG by cytoplasmic esterases [22]. The addition of 10 mM of dimethyl-2-OG induced a significant increase of the reference FRET ratio in cells bearing the F-mT_PII by contrast to cells bearing the F-mT_PII^K58M^ construct, demonstrating that the biosensor responds to 2-OG changes in vivo (Figure 3C). We observed a decrease of the FRET ratio in the control (0 mM dimethyl-2-OG, gold line), likely due to photobleaching induced by the large number of excitation of the fluorophores during the experiment. Next, we followed 2-OG changes in *Anabaena* after the deprivation of combined nitrogen. It is well described that in these conditions, the internal 2-OG concentration will increase to a peak level 1.5 to 2 h after depletion, and then decreases gradually to the basal level [3]. Figure 3D shows that the FRET ratio followed a similar pattern, with an increase at 2 h after nitrogen deprivation, followed by a decrease measured at time 6 h after nitrogen step down. In contrast, we did not detect any variation for cells producing the K58M biosensor variant that indicates that the FRET ratio measured actually reflects 2-OG binding.

For *E. coli*, we used the F-mT-NtcA biosensor. Recombinant *E. coli* strains producing F-mT_NtcA or its R88E mutant were grown in M9 medium supplemented with glucose to the exponential phase. Cells were then transferred into carbon-free M9 medium for 4 h to reduce the intracellular 2-OG level. FRET was then measured over time after addition of 10 mM of glucose or of dimethyl-2-OG. While we did not detect any significant FRET variation in cells depleted of carbon sources, the addition of dimethyl-2-OG or of glucose (which increases 2-OG levels through the TCA cycle) induced a significant increase in the FRET ratio (Figure 3E). This response was due to 2-OG-induced NtcA conformational change, but not to changes in plasmid copy number or biosensor production, as no FRET variation was observed with the mutant version of the biosensor (Figure 3F). The relationship between ammonium abundance in the medium and intracellular 2-OG levels was then examined. It is well described that the presence of nitrogen decreases 2-OG levels. Figure 3G shows that a decrease in ammonium concentration in the medium from 10 mM to 2 mM (Figure 3G, time T1) induces a rapid increase in the FRET ratio, whereas the FRET ratio diminished when ammonium was added at the concentration of 10 mM (Figure 3G, time T2). This opposite correlation between the intracellular concentration of 2-OG and the ammonium level in the medium measured in vivo is consistent with the previous results of 2-OG measurement on cell extracts using HPLC-MS [27] or the PROBS biosensor [35], even if an absolute comparison of the levels obtained by these different methodologies is not accurate. Taken together, we conclude that the two mono-protein biosensors adequately report 2-OG levels variation in vivo.

## 4. Discussion

In this work, we show that PII protein or a domain of NtcA containing the 2-OG binding region could be used as 2-OG FRET mono-protein biosensors when fused to both YFP and CFP/mTurquoise. The two constructs, named F-mT_PII and F-mT_NtcA (with mTurquoise at the C-terminus and YFP midway through the T-loop of PII or close to the N-terminus in NtcA) were based on 2-OG binding dependent conformational changes of the T-loop and of the C-helices, respectively. Variant biosensors, bearing mutations that abolish binding of 2-OG on PII and NtcA, have also been engineered for specificity controls.

We show that these mono-protein biosensors are as efficient as the two-partner based biosensors previously developed for in vitro studies [35,36]. However, these biosensors present additional advantages: by being single polypeptide, only a single purification is required for in vitro studies. They also increase the easy in vivo use as mono-protein biosensors prevent issues regarding differential regulation and/or plasmid copy number when using two constructs. Our results demonstrate that 2-OG binding specifically modifies the FRET ratio of both biosensors, which is not observed with the two mutant versions. One of the most significant properties of the two biosensors tested in our study is their different sensitivity to 2-OG concentrations and therefore their responsiveness to different 2-OG level ranges (Figure 2A,B). This variation of affinity can be inherent to the large difference between the 2-OG binding pockets and the residues involved in 2-OG binding in PII and NtcA [10,12]. 2-OG binding to PII occurs at the micromolar range to fine tune the cellular nitrogen fixation ability as a function of the Carbon/Nitrogen balance [48], whereas 2-OG binds NtcA at high levels during cellular combined nitrogen deprivation [3]. This difference of affinity and binding pocket could explain why the 2-methylenepentanedioic acid (2-MPA) a non-metabolizable analogue of 2-OG, is able to modify the FRET ratio of F-mT_PII but not that of F-mT_NtcA (Figure 2G) [46]. We also noted that the biosensor concentrations did not have any impact on 2-OG level measurements (Figure 2A,B). This observation, as well as the fact that the two biosensors respond to different 2-OG level ranges, represent an advantage to measuring in vivo 2-OG fluctuations in a wide range of organisms. The tools developed here can be therefore applied to many biological systems such as cyanobacteria, bacteria, fungi, plants, and animals.

Importantly, we showed that the two biosensors are able to detect variations of 2-OG levels within minutes (Figure 3A,B), which is particularly crucial when probing 2-OG fluctuations during time-lapse experiments is needed or after a change in the experimental conditions. By modulating glucose, 2-OG and ammonium concentrations in the medium, we observed that the NtcA-based biosensor F-mT_NtcA can detect 2-OG fluctuations in *E. coli* populations as a function of Carbon or Nitrogen availability. Even if our aim is to determine 2-OG variations in vivo and not to quantify the absolute 2-OG concentrations, the data obtained could be considered as consistent with previous results available in the literature. HPLC measurements of the intracellular concentration of 2-OG showed that it increases from 0.35 mM to 2.6 mM after the addition of 10 mM glucose to a carbon-free M9 medium [37]. 2-OG levels have also been shown to significantly decrease after the addition of 10–200 mM NH_4_Cl to an *E. coli* cell culture cultivated under nitrogen starvation conditions [27]. Conversely, *E. coli* cultivated in presence of high ammonium level (10 mM NH_4_Cl) exhibited an increase in intracellular 2-OG from 0.5 mM to 2.5 mM under nitrogen starvation [30]. Regarding the PII-based biosensor, it was found to detect 2-OG levels within the physiological range (10^−2^–10^−5^ M) in vitro (Figure 2A). We further showed that it is efficient to measure 2-OG levels in vivo in *Anabaena* cells (Figure 3C,D). Here again, our data are in agreement with the previously published results using cell extracts [3]. This biosensor is likely to be a powerful tool to precisely measure 2-OG variations in the different stages of heterocyst differentiation. However, PII being a highly conserved protein in bacteria and plants that controls the expression and activity of many enzymes such as the glutamine synthetase, the acetyl-CoA carboxylase or the nitrogenase [14,49,50,51,52,53,54,55], a PII-based 2-OG biosensor is probably not the most appropriate construct to measure 2-OG fluctuations in the majority of living organisms as its overproduction could impact regulatory cascades and cellular activities, not by affecting cellular 2-OG levels but rather by titrating GlnB/PII partners. By contrast, the NtcA regulator is only found within the cyanobacterial phylum. In addition, to avoid any crosstalk in other organisms, the NtcA-based biosensor we constructed lacks the DNA binding domain, and therefore lost its ability to control gene expression in *Anabaena*. This biosensor is therefore most suitable for the study of 2-OG fluctuations in the majority of organisms. 

These biosensors represent the first step towards the development of fluorescent sensing systems for 2-OG able to measure the difference of concentration of this universal signaling molecule in single cells. Two others 2-OG biosensors (PII-TC3 and PII-TC3-R9P) based on the PII skeleton from *Synechococcus elongatus* PCC 7942 have been described recently [56]. Theses 2-OG biosensors have also been constructed on the basis of conformational change of this loop to measure a change in FRET. The PII proteins from *Anabaena* and *S. elongatus* share the same size (112 amino acids) and a high similarity of sequence. However, the structures of these biosensors (PII-TC3 and PII-TC3-R9P) and the one described in our study are not identical. Indeed, in the case of the *Synechococcus* biosensors, the fluorophore donor is inserted in the T-loop, while it is the acceptor fluorophore in the case of *Anabaena* biosensor. Moreover, the acceptor is fused to the C-terminus of PII via the 12-amino-acid streptavidin affinity tag and the fluorophores used are different. Lüddecke et al., fused PII to the circularly permuted mCerulean and the Venus FP, instead of mTurquoise and YFP for F-mT_PII [56]. The most striking difference between the two systems is that the fluorophore is not inserted at exactly the same position in the T-loop. In the case of *Synechococcus* PII-based biosensors, the wild-type sequence results in a too sensitive response. Therefore, a substitution has been introduced in the sequence of PII in order to obtain a relevant biosensor. In the case of the biosensor described here, the wild type sequence of PII used turned out to be compatible with physiological concentrations measure. These structural differences between the two biosensors might explain the fact that the two systems respond to different concentrations of 2-OG. 

Nevertheless, the F-mT_PII and F-mT_NtcA biosensors should be tested to define the best construct for the organism under investigation. In order to further improve the ability of the biosensors to measure the dynamic of 2-OG changes in vivo over extended periods of time, several parameters should be considered. First, as for many fluorophores, repetitive fluorescence excitations yield photobleaching that delays or impedes any time-lapse measurement. We already replaced the CFP donor fluorophore by mTurquoise, which has been shown to provide a higher brightness while being less subject to photobleaching. Future improvements may include switching the YFP by a better-suited fluorophore. Second, in vivo FRET measurements need to mobilize a larger set of controls as biosensor skeletons fused to a single fluorophore, in order to modulate the data in function of fluorescence leakage and heterogeneous levels of production from one cell to another. Nevertheless, the FRET measurement can also be analyzed using alternative methods. Here, we determined the FRET ratio by measuring the intensity of fluorescence of the donor and the acceptor. The FLIM-based FRET technic (Fluorescence-Lifetime Imaging Microscopy) could also report the FRET. These measurements focus on changes in the lifetime of the donor, which prevents some drawbacks of the intensity-based FRET measurement, such as the signal cross-contamination and photobleaching. Each fluorophore has its own characteristic lifetime that is dependent on its molecular environment but independent of its cellular concentration. MTurquoise is a fluorophore that has a better performance in brightness but above all two close lifetimes of 3.7 ns and 3.8 ns that can be regarded as a single lifetime by contrast to CFP that possesses two lifetimes more distanced (2.3 ns and 3.0 ns). Switching the donor fluorophore CFP to mTurquoise circumvents the challenging problem to provide a stable and uniform production of the biosensors and allows measuring in vivo 2-OG levels at the cellular level. However, the FLIM approach could be either an alternative or an additional method of FRET measure.

Finally, additional biosensors based on 2-OG responsive enzymes or regulators should also be engineered and characterized to expand the set of available tools, such as the GOGAT enzyme of *Bacillus subtilis* or *Synechococcus* sp. [57,58] or NrpR transcriptional regulator of *Methanococcus maripaludis* [59,60]. Such tools will not be only important for studying heterocyst differentiation in filamentous cyanobacteria, or the TCA cycle and 2-OG signaling in various organisms in which 2-OG plays important functions, but might provide insights on human diseases that are associated with 2-OG levels imbalance or perturbations, such as acute myeloid leukemia, low-grade glioma, secondary glioblastoma, or that could be cured thanks to 2-OG inhibition of angiogenesis properties [61,62,63,64,65].

## Figures and Tables

**Figure 1 life-08-00051-f001:**
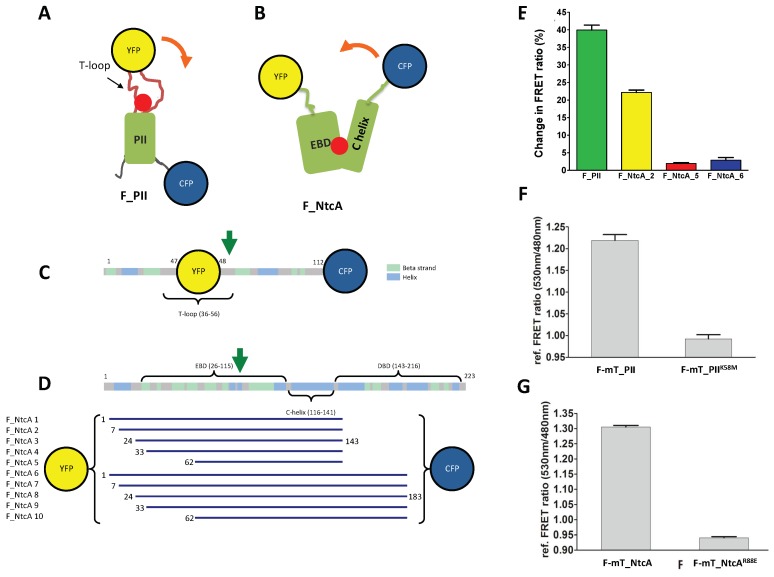
2-oxoglutarate (2-OG) fluorescence resonance energy transfer (FRET) biosensors based on mono-proteins. (**A**,**D**) Schematic (**A**) and linear (**C**) representations of the FRET biosensor F_PII. The CFP is fused at the C-terminal end of PII whereas the YFP is inserted within the flexible, 2-OG-responsive T-loop; (**B**,**D**) schematic (**B**) and linear (**D**) representations of the FRET biosensor F_NtcA. The DNA binding domain was removed and the biosensors have conserved parts of the effector binding domain (EBD) and of the C-helix. The YFP is fused at the N-terminus of the EBD fragments whereas the CFP is fused to the C-terminus of the C-helix fragment. The yellow and blue spheres correspond to YFP and CFP, respectively. The red ball indicates the 2-OG binding site. The orange arrow indicates the conformational changes triggered by 2-OG binding. The green arrows indicate the substituted residues that prevent 2-OG binding and used as controls (K58M for PII, R88E for NtcA). The 10 different NtcA biosensor length variants, highlighting the first and last residues of each construct, are shown in panel (**D**); (**E**) FRET ratio change (%) of the biosensors purified after the addition of 1mM of 2-OG; (**F**,**G**) *In vitro* relative FRET ratio of F-mT_PII and its K58M mutant (**F**) and of F-mT_NtcA_2 and its R88E mutant (**G**) after addition of 10 mM of 2-OG (FRET ratio without 2-OG was set as 1).

**Figure 2 life-08-00051-f002:**
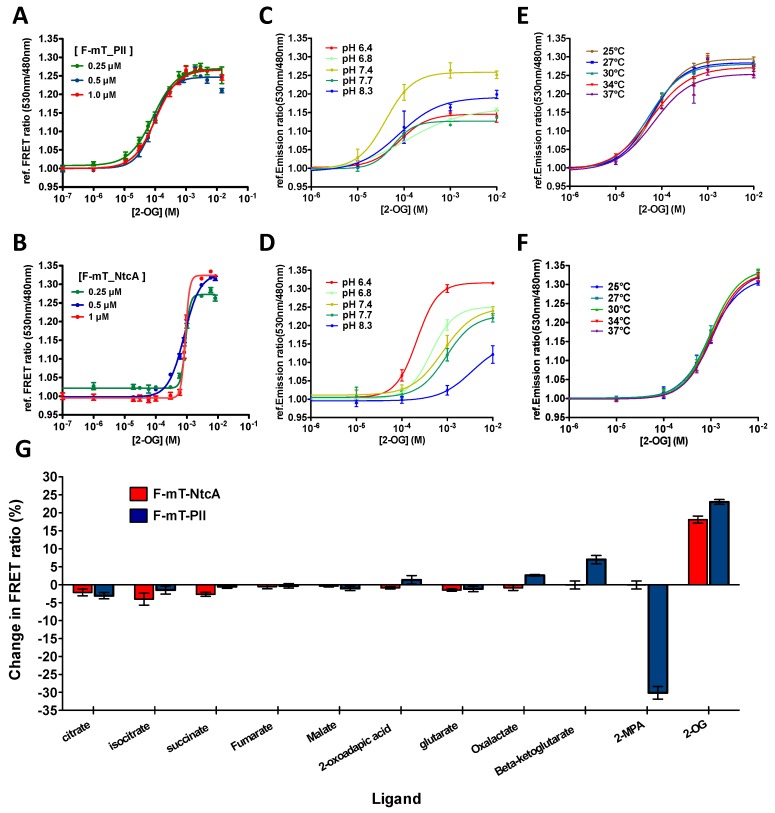
*In vitro* features of the 2-OG FRET biosensors. (**A**,**B**) Relative FRET ratio of F-mT_PII (**A**) and F-mT_NtcA (**B**) measured in vitro as a function of 2-OG concentration. Three different concentrations of proteins were tested (0.25 μM, green line; 0.5 μM, blue line; and 1 μM, red line); (**C**,**D**) 2-OG titration curve of F-mT_PII (**C**) and F-mT_NtcA (**D**) as a function of pH values (from 6.4 to 8.3) with 2-OG concentrations from 10^-6^ M to 10^-2^ M (0.5 μM protein at 30 °C); (**E**,**F**) 2-OG titration curve of F_PII-mT (**E**) and F_NtcA-mT (**F**) as a function of temperature (from 25 °C to 37 °C) with 2-OG concentration from 10^−6^ M to 10^−2^ M (1 μM protein at pH 7.4); (**G**) FRET ratio change (%) of F_NtcA-mT (red) and F_PII-mT (blue) obtained after the addition of 1mM of different metabolites. All results were obtained from three independent measurements.

**Figure 3 life-08-00051-f003:**
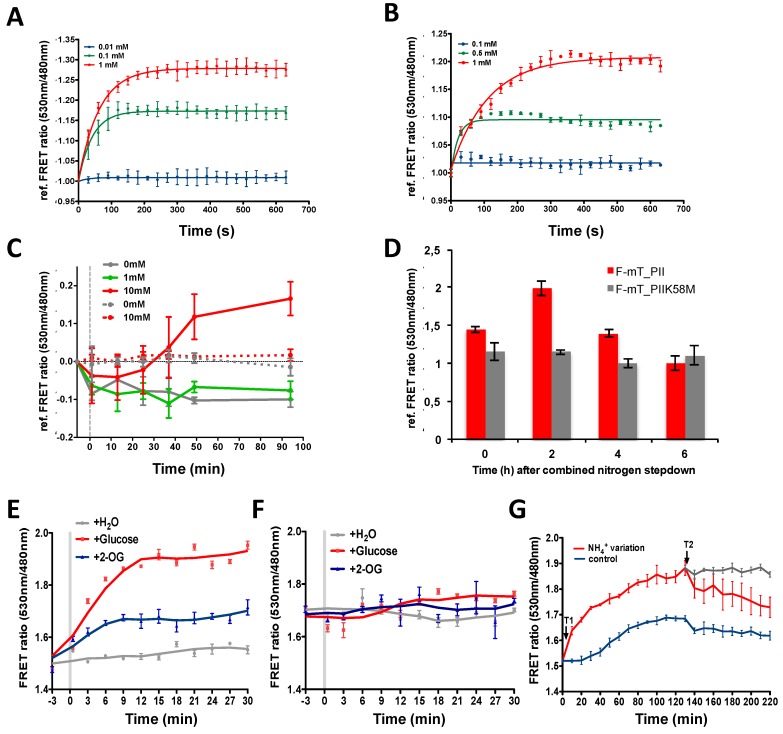
*In vitro* and in vivo features of the 2-OG FRET biosensors. (**A**,**B**) Time-lapse recordings of F-mT_PII (**A**) and F-mT_NtcA (**B**) responses upon addition of different concentrations of 2-OG. The relative FRET ratio was plotted against time (in seconds). The emission data were recorded every 30 s for 11 min with 0.5 μM of protein at 30 °C, pH7.4. Three different concentrations of 2-OG were added (0.01 mM, blue line; 0.1 mM, green line; and 1 mM, red line) for F-mT_PII and (0.1 mM, blue line; 0.5 mM, green line; and 1 mM, red line) for F-mT_NtcA. (**C**) Time-lapse recordings of FRET ratio variations in *Anabaena sp. strain PCC 7120* (*Anabaena*) cells bearing the F-mT_PII 2-OG biosensor (solid lines) or F-mT_PII^K58M^ biosensor (doted lines) before (grey line) or after addition of different concentrations of dimethyl-2-OG (1 mM, green line; 10 mM, red line) in the culture medium; (**D**) Time-lapse recording of FRET ratio variations of *Anabaena* cells bearing the F-mT_PII 2-OG biosensor (red) or F-mT_PII^K58M^ (grey) at different time points after combined-nitrogen step-down. The FRET ratio is referenced as 1 for the lowest value measured for each construction; (**E**,**F**) Time-lapse recordings of FRET ratio variations in *E. coli* bearing 2-OG biosensor F-mT_NtcA (**E**) or F-mT_NtcA^R88E^ (**F**) (grey line), after addition of 10 mM of dimethyl-2-OG (blue line) or 10 mM of glucose (red line) in the culture medium. Data were recorded every 3 min for 30 min; (**G**) Time-lapse recording of the FRET ratio of *E. coli* cells bearing the F-mT_NtcA 2-OG biosensor in presence of 10 mM of ammonium (blue line) or subjected to changes in ammonium concentrations. T1: red line, decrease of ammonium concentration from 10 mM to 2 mM; T2: addition (red line) or no addition (grey line) of ammonium to a final concentration of 10 mM) in the culture medium. Data were recorded every 10 min for 4 h. The experiments were repeated three times, and the mean value and the standard deviation are shown.

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
