# Peer review of "Biosensors-Based In Vivo Quantification of 2-Oxoglutarate in Cyanobacteria and Proteobacteria"

_life, 2018, doi:10.3390/life8040051_

Round 1

Reviewer 1 Report

This manuscript reports a novel FRET-base biosensor of 2-oxoglutarate. This research also showed that this design of biosensor could be used to measure in vivo 2-oxoglutarate level.

1.    Please also provide the result of time-lapse recordings of FRET ratio variations in Anabaena cells bearing the F-mT_PII K58M mutant biosensors after addition of different concentrations of dimethyl-2-OG in the culture medium (A figure similar to Figure 2E).

2.    To figure 2F, please also provide the result when you were using BG110 medium (BG11 without combined nitrogen). It is better to show both BG11 (containing sodium nitrate) or BG110 (BG11 without combined nitrogen).

3.    Please provide the result about the protein purification (such as SDS-PAGE).

4.    In figure 2I, I think it is better also to include another control (FRET ratio of E. coli cells bearing the F-mT_NtcA 2-OG biosensor in the presence of 2 mM of ammonium).

5.    In figure 2E, why the FRET decrease in control (0mM dimethyl-2-OG, gold line). Please explain the possible reason?

6. please check the grammar and English.

Author Response

Itemized response to reviewer's comments

Biosensors-based in vivo quantification of 2-oxoglutarate in Cyanobacteria and Proteobacteria

H.L. Chen, A. Latifi, C.C. Zhang, C.S. Bernard

Life 372549

Reviewer 1

This manuscript reports a novel FRET-base biosensor of 2-oxoglutarate. This research also showed that this design of biosensor could be used to measure in vivo 2-oxoglutarate level.

1.    Please also provide the result of time-lapse recordings of FRET ratio variations in Anabaena cells bearing the F-mT_PII K58M mutant biosensors after addition of different concentrations of dimethyl-2-OG in the culture medium (A figure similar to Figure 2E).

In the revised version of our manuscript, the FRET ratio variations in Anabaena cells bearing the F-mT_PII K58M mutant biosensors are provided in Figure 3C. Please note that, to facilitate the lecture of the figure, the values were presented as ref. FRET ratios.

2.    To figure 2F, please also provide the result when you were using BG110 medium (BG11 without combined nitrogen). It is better to show both BG11 (containing sodium nitrate) or BG110 (BG11 without combined nitrogen).

The information asked by Reviewer 1 is given in Figure 2C for a period of 1h30 minutes. As shown in this figure, the FRET ratio does not increase when the strains are in BG11. We believe that adding 8 more histogram bars, all set to 1 in Figure 3D, would unnecessarily overload the Figure.

3.    Please provide the result about the protein purification (such as SDS-PAGE).

The protein purification asked by Reviewer 1 is provided in the supplementary data Figure S1.

4.    In figure 2I, I think it is better also to include another control (FRET ratio of E. coli cells bearing the F-mT_NtcA 2-OG biosensor in the presence of 2 mM of ammonium).

We agree with Reviewer 1. We added the requested data (2-mM constant concentration of ammonium at T2) in Figure 3G.

5.    In figure 2E, why the FRET decrease in control (0mM dimethyl-2-OG, gold line). Please explain the possible reason?

A decrease of the FRET ratio could be observed in the control (0 mM dimethyl-2-OG, grey line, Figure 3C). The most plausible explanation is that this decrease is due to photobleaching, as the fluorophores are regularly excited during the course of the experiment. We agree with Reviewer 1 that this must be explained. In our revised manuscript this explanation is given in 270

6. Please check the grammar and English.

The text has been verified by a native English speaker.

Reviewer 2 Report

Life 372549 Review

Chen and colleagues presented the development of a pair of biosensors for 2-oxoglutarate based on PII and NtcA. Both candidate constructs were characterized in vitro and in vivo. The narrative of biosensor development is complete but would benefit from a direct comparison to classical analytical methods. The NtcA biosensor appears to be novel but may be limited to qualitative high/low type analysis due to a smaller dynamic range in its current form. Tools such as these may in future be applied as an alternative to HPLC but more development is needed prior to wide adoption of this technology for applications such as cancer research.

Major concerns

1. The authors refer to the description of a PII biosensor with highly similar architecture and properties. The discussion should elaborate in more detail the advancement of this manuscript over Lüddecke et al. 2017.

2. In vitro affinity can be directly inferred from the data presented in Figure 2 A and B but this analysis was not presented. The authors can refer to numerous sources on non-linear regression and fitting a plot of fluorescent signal (including FRET) against ligand concentration.

3. Bioassay is not calibrated or compared to actual in vivo concentrations analyzed by HPLC. In the discussion the authors relate changes in fluorescent intensity to literature values that may or may not be playing out in their system. This speculation should be worded more carefully to avoid the interpretation that any effort was made to tie fluorescent signal to substrate concentration.

Minor concerns

Line 235 Concentration in Anabaena should be stated as it was for E. coli.

In the discussion 324-328 affinity constants were discussed in relation to biological 2-OG which was not measured as opposed to in vitro affinity that was directly measured. This should be changed.

The methods for the ammonium stepdown experiment were not described. How was the concentration of ammonium measured/controlled? Also the data for this experiment shows an identical pattern for the mutant construct. How does one deconvolute which metabolite is responsible for the observed signal without doing HPLC validation?

Fig S1F shows a big decrease for 2-MPA but the identity of 2-MPA does not appear anywhere in the text and an explanation for this phenotype was not provided.

Author Response

Itemized response to reviewer's comments

Biosensors-based in vivo quantification of 2-oxoglutarate in Cyanobacteria and Proteobacteria

H.L. Chen, A. Latifi, C.C. Zhang, C.S. Bernard

Life 372549

Reviewer 2

Chen and colleagues presented the development of a pair of biosensors for 2-oxoglutarate based on PII and NtcA. Both candidate constructs were characterized in vitro and in vivo. The narrative of biosensor development is complete but would benefit from a direct comparison to classical analytical methods. The NtcA biosensor appears to be novel but may be limited to qualitative high/low type analysis due to a smaller dynamic range in its current form. Tools such as these may in future be applied as an alternative to HPLC but more development is needed prior to wide adoption of this technology for applications such as cancer research.

Major concerns

1. The authors refer to the description of a PII biosensor with highly similar architecture and properties. The discussion should elaborate in more detail the advancement of this manuscript over Lüddecke et al. 2017.

As noted by this referee, two additional 2-OG biosensors (PII-TC3 and PII-TC3-R9P) based on the PII skeleton from Synechococcus elongatus PCC7942 are already available (Lüddecke et al., 2017). We agree with the reviewer that the similarities and differences between the different biosensors should be described in further detail. We have therefore amended the discussion section to discuss this point:

Two others 2-OG biosensors (PII-TC3 and PII-TC3-R9P) based on the PII skeleton from Synechococcus elongatus PCC 7942 have been described recently [49]. Theses 2-OG biosensors have also been constructed on the basis of conformational change of this loop to measure a change in FRET. The PII proteins from Anabaena and S. elongatus share the same size (112 amino-acids) and a high similarity of sequence. However, the structures of these biosensors (PII-TC3 and PII-TC3-R9P) and the one described in our study are not identical. Indeed, in the case of the Synechococcus biosensors, the fluorophore donor is inserted in the T-loop, while it is the acceptor fluorophore in the case of Anabaena biosensor. Moreover, the acceptor is fused to the C-terminus of PII via the 12-amino-acid streptavidin affinity tag and the fluorophores used are different. Lüddecke et al., fused PII to the circularly permuted mCerulean and the Venus FP, instead of mTurquoise and YFP for F-mT_PII [49]. The most striking difference between the two systems is that the fluorophore is not inserted at exactly the same position in the T-loop. In the case of Synechococcus PII-based biosensors, the wild-type sequence results in a too sensitive response. Therefore, a substitution has been introduced in the sequence of PII in order to obtain a relevant biosensor. In the case of the biosensor described here, the wild type sequence of PII used turned out to be compatible with physiological concentrations measure. These structural differences between the two biosensors might explain the fact that the two systems respond to different concentrations of 2-OG.

2. In vitro affinity can be directly inferred from the data presented in Figure 2 A and B but this analysis was not presented. The authors can refer to numerous sources on non-linear regression and fitting a plot of fluorescent signal (including FRET) against ligand concentration.

We agree with the reviewer. Thanks to non-linear regression we could determine the dissociation constant of F-mT_PII and F-mT_NtcA to 120 mM and 1,9 mM, respectively. These values are consistent with our results and the Kd of F-mT_PII could be compared to the Kd of 91 mM of the PII-TC3-R9P biosensor previously described by Lüddecke et al.

3. Bioassay is not calibrated or compared to actual in vivo concentrations analyzed by HPLC. In the discussion the authors relate changes in fluorescent intensity to literature values that may or may not be playing out in their system. This speculation should be worded more carefully to avoid the interpretation that any effort was made to tie fluorescent signal to substrate concentration.

We agree with the reviewer. our goal was not to determine exact 2-OG values, but rather to follow 2-OG level changes. We modulate our statement as below:

Even if our aim is to determine 2-OG variations in vivo and not to quantify the absolute 2-OG concentrations, the data obtained could be considered as consistent with previous results available in the literature (Lines 353-355).

Minor concerns

Line 235 Concentration in Anabaena should be stated as it was for E. coli.

This is now indicated line 246.

In the discussion 324-328 affinity constants were discussed in relation to biological 2-OG which was not measured as opposed to in vitro affinity that was directly measured. This should be changed.

We agree with Reviewer 2 that comparing affinities measured in vitro and in vivo is not rigorous. This part of the discussion has been totally changed to avoid this parallel (lines 335 to 342).

The methods for the ammonium stepdown experiment were not described. How was the concentration of ammonium measured/controlled? Also the data for this experiment shows an identical pattern for the mutant construct. How does one deconvolute which metabolite is responsible for the observed signal without doing HPLC validation?

In this revised version of the manuscript, the methodology for nitrogen starvation is explained in Material and Methods section, lines 150-153. To answer Reviewer 2, before transferring the strains to BG110 (medium without any nitrogen source), they are thoroughly washed with medium without nitrate to remove traces of nitrate. In this system, we do not need to use HPLC to follow nitrogen starvation since the development of heterocysts is a good readout for this condition. As long as combined nitrogen is available, heterocyst formation is inhibited.

Fig S1F shows a big decrease for 2-MPA but the identity of 2-MPA does not appear anywhere in the text and an explanation for this phenotype was not provided.

We are grateful to Reviewer 2 to draw our attention to the absence of the information describing 2-MPA. Non-metabolized analogues to 2-OG have been synthesized and used to demonstrate the 2-OG is the signal inducing cell differentiation. This information is now available and the reference of the work cited, lines 248-250.

Reviewer 3 Report

The paper by Hai-lin Chen et al describes the construction and verification of several 2-oxoglutarate (2OG) sensors based on the use of two cyanobacterial proteins, PII and NtcA, known to bind 2OG. Each sensor is based in only one protein to which two fluorescent proteins are fused in such a way that a conformational change triggered by the binding of 2OG is expected to bring those fluorescent proteins closer to each other, thus increasing a FRET signal. Whereas fluorescent protein-based FRET analysis to detect metabolites has been shown before, the use of mono-protein based sensors to determine 2OG levels is a wise choice. The paper clearly describes the development of the sensors and give examples of their use in vitro and in vivo. I have only comments on the presentation of the work.

1.     The authors should re-check all the citations in the text to give accurate references. In many places (too numerous to specify here) a statement in the text is followed by a reference number that takes the reader to a reference in the References list that has nothing to do with that statement. This will be very confusing for a reader interested in following a specific topic and requires that the authors do a detailed correction of all the literature references in this article.

2.     The paper is not long, and I think Fig. S1 could be transferred to the main text. The data presented in this figure are important and the reader will appreciate to have them at hand.

3.     Lines 353-358: I’m not sure whether binding of 2OG by PII could significantly affect the cellular 2OG levels. This would depend on the concentration of the PII protein produced, could this be estimated to be included in this discussion?

Minor points:

4.     Lines 6 & 7: the e-mail address of the corresponding author is given twice.

5.     Line 19: proteobacterium.

6.     Line 20: “cyanobacterium” and “Anabaena sp. strain PCC 7120” (not PCC7120).

7.     Line 41: I suggest that the authors name the cyanobacterium by its complete and correct name here and indicate how it will be referred to in the rest of the paper: “Anabaena sp. strain PCC 7120 (hereafter Anabaena)”.

8.     Line 46: please move the comma in front of “while”: , while the latter…

9.     Line 63: please give references as numbers.

10.  Line 67: an ntcA mutant (not a ntcA mutant).

11.  Line 110: BG11o medium and BG11 medium.

12.  Line 142: constructs.

13.  Line 171: show (not showed).

14.  Line 172: “when bind 2-OG” would read better.

15.  Line 191: please remove “with” and indicated the concentration of 2-OG (1mM).

16.  Line 194: ration?

17.  Line 274: into better than onto?

18.  Line 308: there is some info missing here. Please try: …YFP midway through the T-loop of PII or close to the N-terminus in NtcA).

19.   Line 316: the easy in vivo useas mono-protein…

20.  Line 356: construct (not construction).

21.  Line 383: technic? technique? 

22.  Line 423: some author names are missing from ref. 8.

Author Response

Itemized response to reviewer's comments

Biosensors-based in vivo quantification of 2-oxoglutarate in Cyanobacteria and Proteobacteria

H.L. Chen, A. Latifi, C.C. Zhang, C.S. Bernard

Life 372549

Reviewer 3

1.     The authors should re-check all the citations in the text to give accurate references. In many places (too numerous to specify here) a statement in the text is followed by a reference number that takes the reader to a reference in the References list that has nothing to do with that statement. This will be very confusing for a reader interested in following a specific topic and requires that the authors do a detailed correction of all the literature references in this article.

We thank the reviewer for noticing this. Our apologies for this mistake. It appears that the reference list was not appropriately generated by the software, and we did not see that before submission. This is now fixed in the revised version.

2.     The paper is not long, and I think Fig. S1 could be transferred to the main text. The data presented in this figure are important and the reader will appreciate to have them at hand.

As suggested by this referee, we have added the figures presented in the Fig. S1 to the main text, and reorganized the figures accordingly. (Fig. 1E, Fig 2C,D,E,F,G)

3.     Lines 353-358: I’m not sure whether binding of 2OG by PII could significantly affect the cellular 2OG levels. This would depend on the concentration of the PII protein produced, could this be estimated to be included in this discussion?

We agree with the reviewer. We have modified the discussion accordingly: "A PII-based 2-OG biosensor is probably not the most appropriate construct to measure 2-OG fluctuations in the majority of living organisms as its overproduction could impact regulatory cascades and cellular activities, not by affecting cellular 2-OG levels but rather by titrating its partners."

Minor points:

4.     Lines 6 & 7: the e-mail address of the corresponding author is given twice.

5.     Line 19: proteobacterium.

6.     Line 20: “cyanobacterium” and “Anabaena sp. strain PCC 7120” (not PCC7120).

7.     Line 41: I suggest that the authors name the cyanobacterium by its complete and correct name here and indicate how it will be referred to in the rest of the paper: “Anabaena sp. strain PCC 7120 (hereafter Anabaena)”.

8.     Line 46: please move the comma in front of “while”: , while the latter…

9.     Line 63: please give references as numbers.

10.  Line 67: an ntcA mutant (not a ntcA mutant).

11.  Line 110: BG11o medium and BG11 medium.

12.  Line 142: constructs.

13.  Line 171: show (not showed).

14.  Line 172: “when bind 2-OG” would read better.

15.  Line 191: please remove “with” and indicated the concentration of 2-OG (1mM).

16.  Line 194: ration?

17.  Line 274: into better than onto?

18.  Line 308: there is some info missing here. Please try: …YFP midway through the T-loop of PII or close to the N-terminus in NtcA).

19.   Line 316: the easy in vivo use as mono-protein…

20.  Line 356: construct (not construction).

21.  Line 383: technic? technique?

22.  Line 423: some author names are missing from ref. 8.

We thank the reviewer for the list of grammatical and typographical errors. All of them have been corrected in the revised version.

Round 2

Reviewer 1 Report

The manuscript improves a lot after revision. I am satisfied with the reply of the authors.

1. Check any typing mistake in manuscript again. For instance, line 405 ("bleu line" should be "blue line").

2. Make sure to replace the original figure 2 with the revised version of figure 2.  

Author Response

Itemized response to reviewer's comments 2

Biosensors-based in vivo quantification of 2-oxoglutarate in Cyanobacteria and Proteobacteria

H.L. Chen, A. Latifi, C.C. Zhang, C.S. Bernard

Life 372549

Reviewer 1

1. Check any typing mistake in manuscript again. For instance, line 405 ("bleu line" should be "blue line").

 The text has been verified to remove any typing mistake.

2. Make sure to replace the original figure 2 with the revised version of figure 2. 

We made sure to present the revised version of each figure in the manuscript.

Reviewer 2 Report

The revisions by the authors are acceptable but something happened to the figures during export/uploading that scrambled them. No further suggested revisions at this time.

Author Response

Itemized response to reviewer's comments 2

Biosensors-based in vivo quantification of 2-oxoglutarate in Cyanobacteria and Proteobacteria

H.L. Chen, A. Latifi, C.C. Zhang, C.S. Bernard

Life 372549

Reviewer 2

The revisions by the authors are acceptable but something happened to the figures during export/uploading that scrambled them. No further suggested revisions at this time.

We made sure to present the revised version of the manuscript without mismatch.